# Superfast precipitation of energetic electrons in the radiation belts of the Earth

Xiao-Jia Zhang [1✉], Anton Artemyev [1], Vassilis Angelopoulos [1], Ethan Tsai [1], Colin Wilkins [1], Satoshi Kasahara [2], Didier Mourenas [3], Shoichiro Yokota [4], Kunihiro Keika[2], Tomoaki Hori[5], Yoshizumi Miyoshi [5], Iku Shinohara [6] & Ayako Matsuoka[7]

Energetic electron precipitation from Earth's outer radiation belt heats the upper atmosphere and alters its chemical properties. The precipitating flux intensity, typically modelled using inputs from high-altitude, equatorial spacecraft, dictates the radiation belt's energy contribution to the atmosphere and the strength of space-atmosphere coupling. The classical quasi-linear theory of electron precipitation through moderately fast diffusive interactions with plasma waves predicts that precipitating electron fluxes cannot exceed fluxes of electrons trapped in the radiation belt, setting an apparent upper limit for electron precipitation. Here we show from low-altitude satellite observations, that ~100 keV electron precipitation rates often exceed this apparent upper limit. We demonstrate that such superfast precipitation is caused by nonlinear electron interactions with intense plasma waves, which have not been previously incorporated in radiation belt models. The high occurrence rate of superfast precipitation suggests that it is important for modelling both radiation belt fluxes and space-atmosphere coupling.

[1] Department of Earth, Planetary, and Space Sciences, University of California, Los Angeles, CA 90095, USA. [2] Department of Earth and Planetary Science, School of Science, The University of Tokyo, Tokyo, Japan. [3] Laboratoire Matière en Conditions Extrêmes, Paris-Saclay University, CEA, Bruyères-le-Châtel, France. [4] Department of Earth and Space Science, Graduate School of Science, Osaka University, Toyonaka, Japan. [5] Institute for Space Earth Environmental Research, Nagoya University, Nagoya, Japan. [6] Institute of Space and Astronautical Science, Japan Aerospace Exploration Agency, Sagamihara, Japan. [7] Graduate School of Science, Kyoto University, Kyoto, Japan. ✉email: xjzhang@ucla.edu

Earth's outer radiation belt, a torus-shaped region close to the planet, is filled with energetic electrons[1]. Because electron fluxes increase dramatically during geomagnetic storms, threatening satellites in that region[2], they have been studied theoretically and observationally[3] throughout the entire history of space exploration. These fluxes are controlled by a delicate balance between multiple accelerations and loss processes in Earth's magnetosphere[4]. Whistler waves are particularly important for accelerating electrons to relativistic energies[5–8] and for scattering them in pitch angle, $\alpha$ (the angle between electron speed and magnetic field direction), causing their precipitation into Earth's atmosphere[9–11]. Figure 1a depicts this process of electron scattering by waves and the resulting precipitation. Whistler-wave interaction with electrons has been traditionally modelled in the quasi-linear regime[12], as random jumps in electron energy and pitch angle due to a superposition of low-amplitude, randomly phased wave packets. The random-phase assumption may not be valid for intense wave packets[13], and coherent nonlinear interactions, such as advection and trapping, can lead to a much faster electron transport in pitch angle and energy than in the quasi-linear regime[14–16]. However, such faster nonlinear processes can still transport electrons to small enough pitch angles to eventually reach the dense atmosphere and precipitate, i.e., lose their entire energy through collisions with neutrals. Electrons within the loss cone, the pitch-angle range corresponding to such precipitation (see Fig. 1a), heat the upper atmosphere and alter its chemical properties[17].

There are two opposite limits in quasi-linear diffusion theory[12], as outlined in Fig. 1b. The first corresponds to weak diffusion, in which electrons are slowly scattered into the loss cone on time-scales much longer than the bounce period (the maximum time it can take a loss-cone electron to be lost in the atmosphere), evidenced by a nearly empty loss cone (see the solid grey curve in Fig. 1b showing electron phase space density inside the loss cone, shaded in yellow). It corresponds to weak losses. The second is

strong (or fast) diffusion, in which electrons are scattered into the loss cone so quickly that their atmospheric loss rate simply matches the pitch-angle diffusion rate near the loss cone. In that limit, the electron flux within the loss cone is replenished by diffusion quickly such that it matches the flux just outside the loss cone (see the solid black curve in Fig. 1b). Note that the latter population remains trapped by Earth's magnetic field. It is evident that pitch-angle diffusive processes, by definition, cannot result in loss-cone fluxes that exceed the strong diffusion limit. Thus, in quasi-linear diffusion theory, electron fluxes within the loss cone ($j_{prec}$) will be always lower than, or at most equal to those just outside the loss cone (trapped fluxes, $j_{trap}$).

Here, we demonstrate that, contrary to expectation from classical quasi-linear diffusion theory, precipitation of energetic electrons often occurs faster than prescribed by the strong diffusion limit, exhibiting loss-cone fluxes greater than trapped fluxes. Such a loss-cone overfilling is depicted in Fig. 1c. We show through numerical simulations that this superfast precipitation, observed by low-altitude spacecraft (see Fig. 1a), is caused by electron nonlinear interactions with intense oblique whistler waves measured by conjugate high-altitude spacecraft.

## Results

**Observations.** In Earth's dipole field ($B$), electrons bounce along magnetic field lines while conserving their magnetic moment $\mu = \mathcal{E}\sin^2\alpha/B$, where $\mathcal{E}$ is the electron energy. The loss-cone size (the pitch-angle range of electrons to be lost) is defined as $\sin\alpha_{LC} = \sqrt{B/B_{00}}$ ($B_{00}$ is the magnetic field at the altitude of electron losses). In the outer radiation belt, around the magnetic equator (minimum $B$), $\alpha_{LC}$ is only a few degrees, which makes it challenging for near-equatorial spacecraft to measure electron losses directly[10]. Spacecraft at low altitudes (where $B$ is large and $\alpha_{LC}$ reaches 60–70°), however, can measure electron fluxes within the loss cone. Conjugate spacecraft measurements of near-

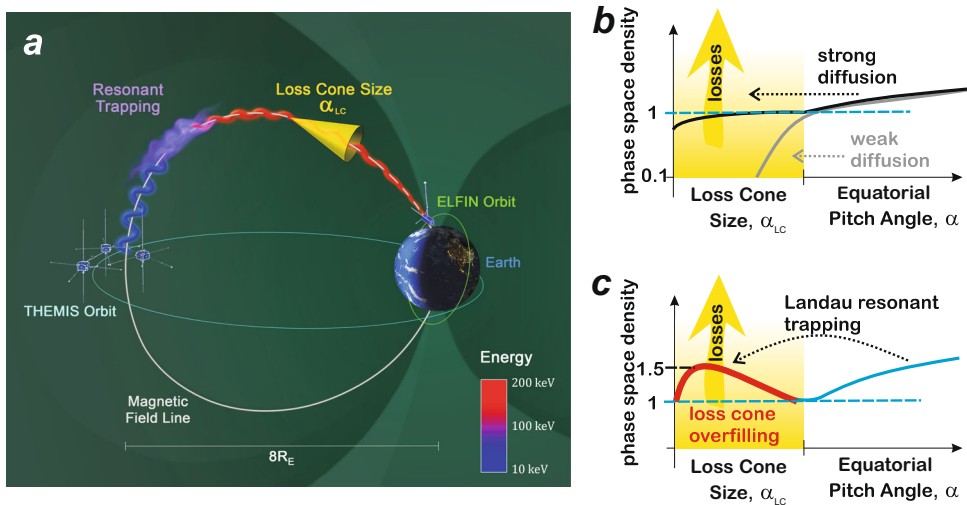

**Fig. 1 Comparison of electron pitch-angle transport caused by wave-driven Landau resonant trapping to transport caused by diffusion. a** Observed precipitation caused by Landau resonant interaction with whistler-mode waves. The initial electron (of approximately 10 keV) has a pitch angle $\alpha$ outside the loss cone, corresponding to a large radius of gyration (blue) around the magnetic field line (white curve). But after resonant trapping and acceleration (to 60–150 keV) by whistler-mode waves (purple) observed at Time History of Events and Macroscale Interactions during Substorms (THEMIS) spacecraft on a near-equatorial orbit (cyan curve), the electron's pitch angle enters the loss cone (yellow) and it is precipitated (red) toward the atmosphere, as observed by Electron Losses and Fields Investigation (ELFIN) spacecraft on a low-altitude polar orbit (green curve). **b** Normalized electron phase space density profiles (solid curves) from within the loss cone (yellow range, of maximum pitch angle $\alpha_{LC}$) to outside it, for weak (grey) and strong (black) diffusion by waves. The electron phase space density within the loss cone remains much smaller than (for weak diffusion) or at most equal to (for strong diffusion) the phase space density of trapped electrons immediately outside the loss cone. **c** Same as (**b**) but for Landau resonant nonlinear trapping by intense oblique waves, leading to loss cone overfilling (red) with higher electron phase space density (solid curve) within the loss cone than immediately outside it, and faster electron losses than for strong diffusion.

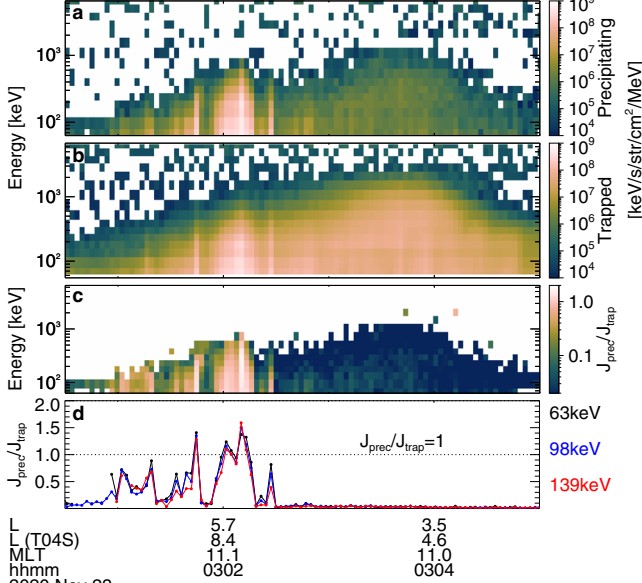

**Fig. 2 Overview of ELFIN-A spacecraft observations on 23 November 2020.** Energy spectra of **a** precipitating $j_{prec}(E)$ and **b** trapped $j_{trap}(E)$ electron fluxes, **c** precipitating to trapped flux ratio $j_{prec}/j_{trap}$, **d** $j_{prec}/j_{trap}$ at 63 keV, 98 keV, and 139 keV, with the dotted line showing $j_{prec}/j_{trap} = 1$. ELFIN-A's $L$ location from dipole and TSO4[69] magnetic field models ($L$ is the equatorial geocentric distance of the magnetic field line, in Earth radii) and magnetic local time ($MLT$) are shown in the bottom.

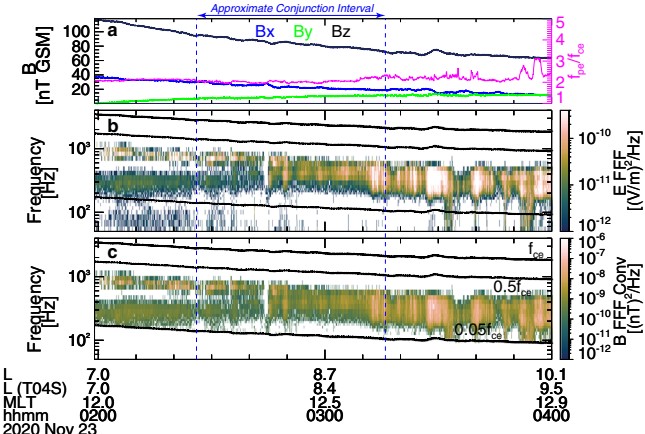

**Fig. 3 Overview of THEMIS-E spacecraft observations from 02:00 to 04:00 UT on 23 November 2020 (the interval of approximate conjunction with ELFIN-A spacecraft is delimited by two vertical dashed blue lines). a** Magnetic field vector (left axis): $B_x$ (blue), $B_y$ (green), and $B_z$ (black) components in the geocentric solar magnetospheric (GSM) coordinate system, and plasma frequency to gyrofrequency ratio (magenta) $f_{pe}/f_{ce}$ (right axis), whistler-mode wave **b** electric field spectra and **c** magnetic field spectra converted from electric spectra. THEMIS-E $L$ location from dipole and TSO4 magnetic field models ($L$ is the equatorial geocentric distance of the magnetic field line, in Earth radii) and magnetic local time ($MLT$) are shown in the bottom.

equatorial waves responsible for electron scattering into the loss cone and low-altitude electron fluxes within it, are needed to quantify electron precipitation rates[18]. We use such a combination of observations from the equatorial Time History of Events and Macroscale Interactions during Substorms (THEMIS) spacecraft (see ref. [19] and "Methods", subsection "Modelling

technique") and the low-altitude Electron Losses and Fields Investigation (ELFIN) spacecraft[20]—see Fig. 1a. ELFIN consists of two spinning CubeSats (ELFIN-A/B) that provide high-resolution pitch-angle- and energy-resolved measurements of energetic electron fluxes at all latitudes at altitudes of about 400–450 km.

Figures 2 and 3 show an overview of an event with conjugate measurements from ELFIN and THEMIS (probe E). From approximately 03:01–03:04 UT on 23 November 2020, ELFIN crossed the outer radiation belt (indicated by a $j_{trap}$ increase up to 1 MeV in Fig. 2a) in the southern hemisphere and observed very strong electron precipitation ($j_{prec} \sim j_{trap}$ for <200 keV in Fig. 2b) at $L \sim 7.5$–10 (distance from Earth in Earth radii). At the same time, THEMIS was also at $L \sim 7.5$–10, in the same noon magnetic local time (MLT) sector as ELFIN, very close to the magnetic equator (where the magnetic field is dominated by GSM $B_z$, as in Fig. 3a). It is worth emphasizing that THEMIS continuously observed strong whistler-mode waves with frequencies $f \in [0.1, 0.3]$ of the electron gyrofrequency ($f_{ce}$, as in Fig. 3b, c) from 02:00 UT to 04:00 UT. Prior THEMIS statistics have shown that the typical correlation length of the source region of whistler-mode waves is about 1.4 h in MLT and 1.5 Earth radii radially at $L \sim 7.5$–10 near 11–12 MLT[21]. Accordingly, an approximate conjunction (within 1.4 MLT, 1.5 in $L$, and 35 min UT) between the observed ELFIN precipitation and THEMIS waves occurred between 02:26 UT and 03:16 UT in Fig. 3. Therefore, it is reasonable to assume that the precipitation detected by ELFIN at 03:02 UT is driven by the whistler-mode waves measured by THEMIS near 03:00 UT in Fig. 3 (as well as near 02:30 UT in Fig. 4). According to ELFIN measurements, $j_{prec}$ not only reached $j_{trap}$, but actually exceeded it for electron energies <200 keV, see Fig. 2d. Because such strong electron precipitation cannot be explained by the classical pitch-angle diffusion theory, we seek an alternate mechanism that could be responsible for it.

Measurements from THEMIS show that the observed whistler-mode waves carry strong field-aligned electric fields and have an elliptical polarization (see Fig. 4). These are typical properties of waves propagating at very large angles relative to the background magnetic field[22,23]. Such very oblique waves have been previously observed in the same ($L, MLT$) range by THEMIS[24], and at similarly low frequencies by Van Allen Probes[25], although they are more common at higher frequencies[26]. Such oblique waves may interact with electrons through Landau resonance when the electron field-aligned velocity is equal to the wave phase velocity (ratio of wave frequency to wavenumber), $v_{\parallel} = 2\pi f/k_{\parallel}$. Oblique whistler-mode waves are not as common as the most intense field-aligned whistler-mode waves[25,27]. Specific plasma conditions are required for these oblique waves to survive the strong damping caused by suprathermal electrons: the electron distribution function must have a reduced gradient within the $v_{\parallel}$ range corresponding to equatorial Landau resonance (see ref. [23] and references therein). Such an electron distribution, commonly observed on the dayside, is also present in our event (see "Methods", subsection "Statistics of precipitation with loss-cone overfilling"). In addition, the observed oblique whistler-mode waves are sufficiently intense (with electric field amplitude reaching approximately $10 \text{ mV m}^{-1}$, see examples of wave packets in Fig. 4) to trap electrons in the wave potential via Landau resonance[23,28]. Such intense waves can therefore interact with electrons nonlinearly, leading to a fast acceleration and pitch-angle decrease of phase-trapped electrons[15,22]. Landau trapping by oblique waves differs from the commonly investigated trapping by field-aligned waves (those waves move accelerated electrons away from the loss cone[29], not into the loss cone as oblique ones do in our case). The combined electron acceleration (~10–30 keV electrons are accelerated to ~100 keV)

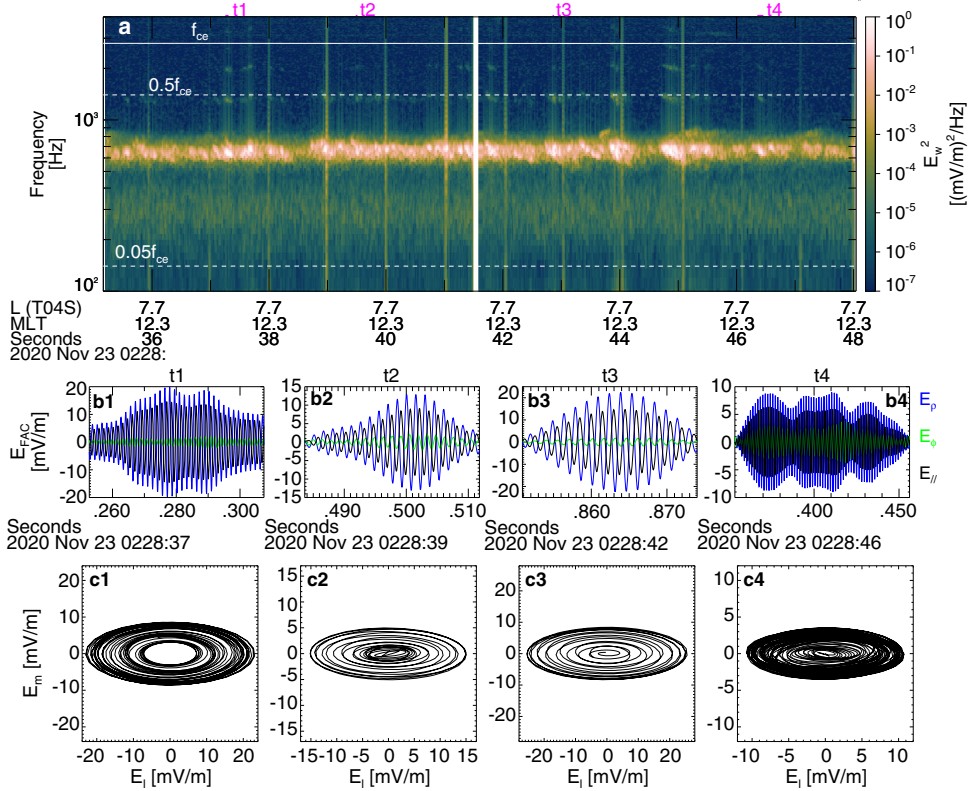

**Fig. 4 Whistler-mode wave characteristics as observed by THEMIS-E spacecraft. a** Whistler-mode wave electric field spectra, four examples (at times corresponding to t1–t4 denoted by magenta bars in panel a) of wave packets with **b** electric fields in local geomagnetic-field-aligned coordinates and **c** electric field hodograms of intermediate variance component ($E_m$) versus maximum component ($E_l$).

and pitch-angle decrease by Landau trapping can result in a large increase of the electron flux within the loss cone (Fig. 1d) due to the large phase space density of the 10–30 keV source electrons, thus explaining the loss-cone overfilling ($j_{prec} > j_{trap}$) observed by ELFIN in Fig. 2d.

**Comparison with numerical simulations**. Next, we performed numerical simulations to verify our hypothesis that the loss-cone overfilling is caused by nonlinear Landau resonance of ~10–100 keV electrons with oblique whistler-mode waves. We combined the observed equatorial electron spectrum, wave intensities, and frequencies from THEMIS to evaluate the evolution of the distribution in phase space[30] and derive the expected electron energy and pitch-angle distribution at ELFIN (see "Methods", subsection "Modelling technique"). Figure 5 shows a comparison of electron pitch-angle distributions observed by ELFIN with those obtained from numerical simulations. ELFIN measurements show a clear increase in electron fluxes at about 100 keV from the trapped pitch-angle range ($90° < \alpha < \alpha_{LC} \simeq 115°$) to the precipitating pitch-angle range ($\alpha > \alpha_{LC}$). That this pattern of pitch-angle distributions is observed during multiple spin periods of ELFIN implies that this observation is not due to time aliasing, and that loss-cone overfilling persists for long temporal (3–10 s) and spatial scales (20–100 km in the ionosphere).

Landau trapping can result in a rapid equatorial pitch-angle reduction from $\alpha_{eq} \sim 4$–$10°$ to $\alpha_{eq} \sim 1.5° < \alpha_{eq,LC}$ during a single resonant interaction: waves trap electrons and quickly transport them directly into the loss cone rather than slowly diffusing them toward it during multiple scatterings with waves[23]. This trapping is accompanied by an energy increase that can be modelled by magnetic moment conservation of electrons in Landau resonance,

$\mathcal{E}\sin^2\alpha_{eq} = $ const.   and   $\Delta\mathcal{E}/\mathcal{E} \sim (\sin\alpha_{eq}/\sin\alpha_{eq,LC})^2 \gg 1$   for initial $\sin\alpha_{eq} > 2\sin\alpha_{eq,LC}$. Trapped electrons can typically gain an energy of tens of keV before being released from the resonance into the loss cone (see "Methods", subsection "Modelling technique", and Supplementary Fig. 6). Therefore, the loss-cone overfilling with approximately 100 keV electrons observed by ELFIN likely results from a nonlinear acceleration of electrons with initial energies of about 10–35 keV around the equator. Because of the slightly higher trapping probability associated with the lower range of initial energies, more electrons are released from trapping at smaller equatorial pitch angles (deeper into the loss cone). This effect of higher fluxes at lower pitch angles is further enhanced by the presence of a flat initial (pre-accelerated) pitch-angle distribution $\sim \sin^{1/6}(\alpha_{eq})$, in the near-equatorial THEMIS observations. The latitudinal confinement of oblique whistler-mode waves[22,23] limits the efficiency of Landau trapping acceleration: ~10 keV electrons trapped near the equator and released from trapping at a higher latitude can gain at most 100–200 keV. And, indeed, pitch-angle distributions measured by ELFIN and those obtained from numerical simulations do not show any loss-cone overfilling ($j_{prec} > j_{trap}$) feature above 300 keV (see Supplementary Fig. 7).

A peculiarity of Landau trapping acceleration is that the final electron energy increases with latitude, in both theory and simulations[15,23]. Near the equator, this overfilling effect is therefore expected to be present only at lower energies, below 30–40 keV. Equatorial observations of electron fluxes within the loss cone have been impossible in the past due to the small loss-cone size at the equator. Recently, however, the Exploration of energization and Radiation in Geospace (ERG/Arase[31]) spacecraft has enabled such observations in the inner magnetosphere[10]. During the event shown in Figs. 2 and 3, ERG, which was in the dayside outer

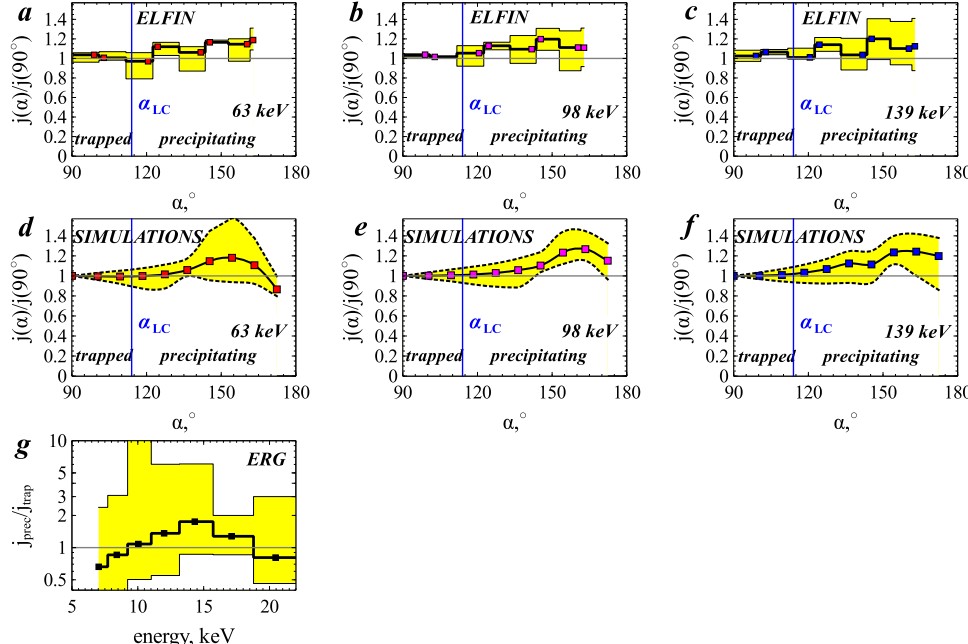

**Fig. 5 Comparison of precipitating electron fluxes (in the loss cone) from observations and from numerical simulations. a–c** Local pitch-angle ($\alpha$) spectra of 63, 98, and 139 keV electron fluxes measured by ELFIN spacecraft, averaged over the most intense precipitation interval (from 03:02:01 to 03:02:14 UT), and normalized to the $\alpha = 90°$ flux average, $j(\alpha)/j(90°)$. **d–f** $j(\alpha)/j(90°)$ from numerical simulations of nonlinear electron Landau resonant interaction with oblique whistler-mode waves measured near the equator by THEMIS-E spacecraft. **g** Precipitating to trapped electron flux ratio $j_{prec}/j_{trap}$ as a function of electron energy measured by Exploration of energization and Radiation in Geospace (ERG/Arase) spacecraft from 02:30:00 to 03:00:00 UT. Yellow regions indicate the highest and lowest flux ratios during the averaging interval (note that for ERG, the highest flux ratios at 12–14 keV have poor counting statistics). Blue vertical lines in the pitch angle spectra indicate the local loss-cone angle, $\alpha_{LC}$, which separates the trapped and precipitating electrons.

radiation belt, detected electron fluxes (see instrument details in[32]) encompassing the loss cone (see Fig. 5g). The strong precipitation at ERG with $j_{prec} \geq j_{trap}$ at 10–20 keV, an energy range at which electrons resonate with oblique whistler-mode waves at the equator, further supports our hypothesis of loss-cone overfilling by nonlinear Landau resonant acceleration (note that to increase the low counting statistics, we averaged ERG measurements for about 20 min; zero count data are omitted from the average because they correspond to times when there are no whistler-mode waves). These electrons were accelerated by trapping near the equator (ERG was below ~25° of magnetic latitude). The further acceleration up to 60–140 keV observed by ELFIN at high latitudes and reproduced in our simulations, however, requires electron trapping up to mid-latitude ($\in [30, 40]°$)[15,23]. Therefore, even when loss-cone-resolving measurements are used aboard near-equatorial spacecraft, such as ERG, they can only observe <20 keV precipitating fluxes from this effect.

## Discussion

Our analysis of strong precipitation measured by ELFIN demonstrates that these observations are associated with THEMIS equatorial measurements of whistler-mode waves. Assuming that these waves are propagating very obliquely to the background magnetic field (THEMIS measurements in the same region 30 min before ELFIN's orbit definitely show such very oblique waves), we show that such oblique whistler-mode waves can significantly enhance electron losses and create strong fluxes of ~100 keV electrons precipitating into the atmosphere. Although these oblique waves are fairly common[25,27] and their potential effect on precipitation has been discussed[23,33], they have been excluded from most radiation belt models due to their low average magnetic field intensity[34]. So how often can these oblique waves produce loss-

cone overfilling for ~100 keV electrons? To address this question, we examine five months of ELFIN observations in the dayside inner magnetosphere, where oblique whistler-mode waves are most common[27]. We find that the occurrence rate of loss-cone overfilling (with $j_{prec}/j_{trap} > 1$) is 10% of all precipitation events (those with $j_{prec}/j_{trap} > 0.05$) at $L \in [10, 12]$. We have also found that during the several fortuitous conjunctions with THEMIS, the loss-cone overfilling observations are associated with very oblique whistler-mode waves on the equator, consistent with the case study reported herein. The occurrence rate remains as high as 5% at $L \in [6, 9]$, and decreases further inward of that distance, down to near-zero at $L < 4$ (see "Methods", subsection "Statistics of precipitation with loss-cone overfilling"). Because loss-cone overfilling corresponds to much stronger electron losses than at other times, an occurrence rate of even 5 to 10% makes it an important contributor to ~50–200 keV electron precipitation. Such strong losses can suppress the source electron fluxes in the ~10–30 keV range and prevent them from acting as a seed population for relativistic energies. Additionally, precipitating ~100 keV electrons can penetrate the atmosphere down to an altitude of about 75 km, where they can significantly alter atmospheric properties and even influence local winter climate[17]. Therefore, the reported effect of loss-cone overfilling is likely important for both radiation belt dynamics and magnetosphere-atmosphere coupling.

## Methods

**Statistics of precipitation with loss-cone overfilling**. Figure 1 in the main text shows a typical loss-cone overfilling event observed by ELFIN, with $j_{prec} > j_{trap}$. From five months of ELFIN observations in the dayside magnetosphere (L-shell below 12, MLT=9-15), we identified 166 orbits with such events (among a total of 465 orbits) and a total number of 943 ELFIN spins with $j_{prec} > j_{trap}$. Supplementary Figure 1 shows that the occurrence rate of loss-cone overfilling can be significant over a wide L-shell range and for sufficiently intense trapped fluxes >10⁶ keV cm⁻² s⁻¹ sr⁻¹ MeV⁻¹.

The observed statistics of loss-cone overfilling are most likely due to electron Landau resonance with very oblique whistler-mode waves, because the more investigated cyclotron resonance cannot explain the formation of the $j_{prec} > j_{trap}$ feature, despite the fact that low-frequency waves present at high $L = 8–10$ in the dawn-noon sector where a plasma frequency to gyrofrequency ratio of $f_{pe}/f_{ce} \sim 5$ can allow cyclotron resonance with >40 keV electrons there[35,36]. To show this, let us consider resonance curves in (energy, pitch-angle) space: the wave-particle resonant interactions move electrons along such curves[37,38]:

$$(\gamma^2 - 1)\sin^2\alpha_{eq} = \text{const.} \quad (1)$$

for Landau resonance (conservation of the magnetic moment) and

$$\left(\frac{2\Omega_{ce,eq}}{\omega}\right)\gamma - (\gamma^2 - 1)\sin^2\alpha_{eq} = \text{const.} \quad (2)$$

for cyclotron resonance, where $\omega$ is the wave frequency, $\Omega_{ce,eq}$ the equatorial gyrofrequency, and $\gamma$ the Lorentz factor. The cyclotron phase trapping may change the electron energy significantly and create a local gradient in the (energy, pitch-angle) space, but for the energy range of interest (<300 keV) the cyclotron phase trapping results in a pitch-angle increase, whereas electron losses occur primarily due to phase bunching[38], which is associated with an energy decrease. Moreover, phase trapping is stronger at lower energies[39], which would produce stronger peaks of $J_{prec}/J_{trap}$ at lower energies, contrary to the present ELFIN observations. Cyclotron phase bunching will move particles gradually toward the loss cone with an energy loss, but it cannot create new gradients in pitch-angle space, because to reach a certain final (precipitating) energy, electrons at smaller pitch angles should start with a larger energy and would therefore result in a smaller flux. In contrast to cyclotron resonance, Landau resonance curves have a very strong gradient around the loss cone ($\sim 1/\sin^2\alpha_{eq}$), i.e., all electrons moving toward small pitch angles should gain energy. Thus, the Landau trapping responsible for pitch-angle decrease can move low-energy (associated with the large fluxes) electrons from moderate pitch angles into the loss cone, and this transport will be associated with an energy gain. The Landau resonance is effective only for quite oblique whistler-mode waves[23], and to further support our conjecture that the loss-cone overfilling is due to Landau trapping, we check wave observations for events in Supplementary Fig. 1 with THEMIS conjunctions. Supplementary Figures 2–4 show three examples (in addition to the event shown in the main text) where ELFIN observations of loss-cone overfilling are associated with THEMIS conjugate observations of oblique (elliptically polarized and having large field-aligned electric field) waves.

Note that the radial distribution of events in Supplementary Fig. 1 further supports the key role of the Landau resonance. As the Landau resonant energy quickly increases for lower background plasma density, ($\mathcal{E}_{res} \sim (\omega/k_\parallel)^2 \sim f_{pe}^{-2}$), loss-cone overfilling for higher energy (>100 keV) electrons is more often observed at higher $L$, where the plasma density is lower. Although here we restricted our dataset to the dayside, to exclude effects of isotropic electron precipitation from the plasma sheet[40], ELFIN captures loss-cone overfilling on the nightside as well. On the nightside, however, strongly nonlinear electrostatic solitary waves (so-called time domain structures consisting of electron holes, electron-acoustic solitons, and double layers[41,42]) may provide the same nonlinear Landau trapping for <300 keV electrons[43] as very oblique whistler-mode waves. Therefore, this phenomenon of superfast losses of ~100 keV electrons can affect a wide range of local times and can be responsible for rapid electron losses from the plasma sheet injection region associated with intense whistler-mode waves and time domain structures[44].

ELFIN data and THEMIS data have been loaded and analyzed using plug-ins to the Space Physics Environment Data Analysis Software (SPEDAS) framework.

**ERG measurements**. We use ERG medium-energy particle experiments-electron analyzer (MEP-e) measurements of $j_{prec}/j_{trap}$ in Fig. 5g. The precipitating flux $j_{prec}$ is calculated based on the electron count during the time period when one of the MEP-e scopes captures the loss cone (time step = 15.6 ms for an energy bin, see. ref. [32]). To determine the trapped flux $j_{trap}$, we use the pitch-angle range of [5, 15]°. This pitch angle range can be scanned multiple times by multiple telescopes during an ERG spin (8 s). The average of these multiple flux values is used to calculate $j_{trap}$. ERG data have been analyzed using plug-ins to the SPEDAS framework.

**Modelling technique**. ELFIN low-altitude observations of loss-cone overfilling are obtained in conjunction with high-altitude equatorial observations of electrons and whistler-mode waves at THEMIS-E. These waves, almost electrostatic, propagate in a very oblique whistler mode[45] and may survive strong Landau damping by suprathermal electrons because of peculiarities in the electron distribution function[46]. And, indeed, the THEMIS Electrostatic Analyzer[47] clearly shows a plateau in both parallel (pitch angles < 30°) and perpendicular (pitch angles ∈ [75, 105]°) directions over the entire wave interval (see Supplementary Fig. 5a–c). The energies of this enhanced electron population (approximately 1–5 keV) are close to the Landau resonant energy at the equator for the observed wave frequencies and wave-normal angle. Such electron distribution functions with reduced velocity gradient in the vicinity of Landau resonant energies significantly reduce Landau damping and provide favourable conditions for the generation of very-oblique whistler-mode waves[23,48].

To simulate wave-particle interactions, we used combined measurements from the Electrostatic Analyzer (<25 keV) and the Solid State Telescope ($\in$ [50, 500] keV) with an interpolation over the energy gap[49]. We obtained the distribution of wave characteristics $\mathcal{P}(B_w, \omega)$ from THEMIS $fff$ wave spectra[50], as measured by electric field antennas[51] and converted them to wave magnetic field using the cold plasma dispersion relation[52] (as calibration of the search-coil measurements for this recent date has not yet taken place). Supplementary Figure 5d shows the distribution of $\mathcal{P}(B_w, f/f_{ce})$ collected during the THEMIS-ELFIN near-conjunction interval. Electric to magnetic field spectral power conversion and simulation of wave-particle interactions also require information about wave normal angles, $\theta$. Using THEMIS electric field wave-form data (see Fig. 4), we estimated $\theta$ to be large, with $\theta \in [\theta_r - 10°, \theta_r - 5°]$ (where $\theta_r$ is the resonance cone angle) for these very oblique whistler-mode waves[53]; these estimates are consistent with general whistler-mode statistics showing the existence of two well-separated whistler-mode wave branches: quasi-parallel whistler-mode waves with $\theta$ below the Gendrin angle and very oblique whistler-mode waves with $\theta$ near the resonance cone angle[25]. In simulations, we therefore use a uniform $\theta$ distribution within the $[\theta_r - 10°, \theta_r - 5°]$ range. Although Supplementary Fig. 5d contains slightly lower $f/f_{ce}$ values than Fig. 4, note that the Landau resonant parallel electron energy $\mathcal{E}_{res}$, most important in simulations, can be written as

$$\mathcal{E}_{res} \simeq 250 \left(\frac{f_{ce}}{f_{pe}}\right)^2 \left(\frac{\cos\theta_r}{\cos\theta}\right)\left(1 - \frac{\cos\theta_r}{\cos\theta}\right) \text{keV}, \quad (3)$$

showing that it is nearly independent of $f/f_{ce}$ for $0.14 < f/f_{ce} < 0.25$ in this $\theta$ range.

We use the mapping technique[30] to test the hypothesis that nonlinear Landau resonance is responsible for the loss-cone overfilling observed by ELFIN. This technique is based on the theoretical model of nonlinear wave-particle interactions characterized by three main parameters: the probability of trapping into the resonance $\Pi(\mathcal{E}, \alpha_{eq})$, the energy change due to trapping $\Delta\mathcal{E}_{trap}(\mathcal{E}, \alpha_{eq})$, and the energy change due to nonlinear scattering $\Delta\mathcal{E}_{scat}(\mathcal{E}, \alpha_{eq})$. The probability of trapping determines the ratio of trapped particles to the total number of particles passing through a single resonance (see analytical theory for $\Pi$ in[54–56]). The stochastization of the wave phase $\xi$ between resonances[57,58] makes individual nonlinear resonant interactions independent[59]. Thus, each electron trajectory can be traced in time as: $t \to t + \tau_{bounce}/2$, $\mathcal{E} \to \mathcal{E} + \Delta\mathcal{E}$, where $\Delta\mathcal{E} = \Delta\mathcal{E}_{trap}$ for $\xi \in 2\pi[0, \Pi)$ and $\Delta\mathcal{E} = \Delta\mathcal{E}_{scat}$ for $\xi \in 2\pi[\Pi, 1]$, and $\xi$ is given by a random uniform distribution $\in [0, 1]$. The corresponding pitch-angle changes are calculated from the conservation of the magnetic moment $\mathcal{E}\sin^2\alpha_{eq}/B_{eq}$ in the Landau resonance. This approach has been developed and verified against test particle simulations in[30,60], whereas analytical equations for $\Pi(\mathcal{E}, \alpha_{eq})$, $\Delta\mathcal{E}_{trap}(\mathcal{E}, \alpha_{eq})$, $\Delta\mathcal{E}_{scat}(\mathcal{E}, \alpha_{eq})$ have been derived and verified against test particle simulations in[61,62]. Five main characteristics of the nonlinear wave-particle interaction ($\Pi$, $\Delta\mathcal{E}_{trap,scat}$, $\Delta\alpha_{trap,scat}$) depend on the preresonance energy and pitch angle ($\mathcal{E}, \alpha_{eq}$), the wave characteristics (frequency $\omega$, wave-normal angle $\theta(s)$ and wave amplitude $B_w(s)$ profiles along magnetic field lines), and background plasma/magnetic field characteristics $\omega_{pe}/\Omega_{ce}$, $B(s)$. We use the distribution of observed wave equatorial characteristics $\mathcal{P}(B_w, f/f_{ce})$ (see Supplementary Fig. 5d) dipole magnetic field model, and several equatorial values of $f_{pe}/f_{ce}$ within the observed range (see Fig. 3a). The profile of $f_{pe}$ along magnetic field lines is adopted from the[63] model, and the profile of $B_w(s)$ for oblique whistler-mode waves is from the[22] model. The wave-normal angle variation along magnetic field lines is set at a constant deviation from the resonance cone angle, $\cos(\theta) = qf/f_{ce}$ (according to statistical results from[53], with ten $q$ such that $\theta$ is uniformly distributed over $[\theta_r - 10°, \theta_r - 5°]$ at the equator. Wave electromagnetic field components are obtained from the cold plasma dispersion relation[64] for given $B_w(s)$, $\theta(s)$, $f$, and $f_{pe}/f_{ce}$. For each resonant interaction (half the bounce period), we use the observed $\mathcal{P}(B_w, f/f_{ce}, \theta)$ distribution and the uniform $q$ distribution to select wave characteristics, and then use $\Pi$, $\Delta\mathcal{E}_{trap,scat}$, $\Delta\alpha_{trap,scat}$ precalculated for all wave characteristics. Such mapping repeats for $10^9$ orbits for $t \in [0, 10]$ min[30] of wave activity with several bursts of observed intense waves (the temporal profile of wave intensity is derived from THEMIS observations around the moment of conjunction with ELFIN). Then these orbits are used to transform the initial energy and pitch-angle distribution (from THEMIS equatorial measurements) into the final distributions mapped at ELFIN's altitude.

The characteristics of wave-particle nonlinear interactions ($\Pi$, $\Delta\mathcal{E}_{trap,scat}$, $\Delta\alpha_{trap,scat}$) cannot be averaged over wave characteristics (as is traditionally done for diffusion rates in quasi-linear theory, see[65]), because each nonlinear trapping can change the electron energy by a magnitude of about their initial energy ($\Delta\mathcal{E}_{trap} \sim \mathcal{E}$). Thus, the mapping technique is based on a probabilistic approach operating with an ensemble of $\Pi$, $\Delta\mathcal{E}_{trap,scat}$, $\Delta\alpha_{trap,scat}$ distributions in ($\mathcal{E}, \alpha_{eq}$) space. Supplementary Fig. 6 illustrates these distributions for the observed $\mathcal{P}(B_w, f/f_{ce}, \theta)$:

$$\Pi^* = \sum_{k,l,j} \Pi_{k,l,j}\mathcal{P}\left(B_{w,k}, f_l/f_{ce}, \theta_j\right), \quad (4)$$

$$\Delta\mathcal{E}_{trap} = \sum_{k,l,k} \Delta\mathcal{E}_{trap,k,l,j}\Pi_{k,l,j}\mathcal{P}\left(B_{w,k}, f_l/f_{ce}, \theta_j\right)/\Pi^*, \quad (5)$$

$$\Delta \mathcal{E}_{scat} = \sum_{k,l,k} \Delta \mathcal{E}_{scat,k,l,j} \mathcal{P}\left(B_{w,k}, f_l/f_{ce}, \theta_j\right). \qquad (6)$$

Low-energy electrons have a higher probability of being trapped into Landau resonance and transported to the low pitch-angle range ($\Delta \alpha_{trap} < 0$), with significant acceleration ($\Delta \mathcal{E}_{trap} > 0$); see three example trajectories from the mapping technique in Supplementary Fig. 6f. These trajectories contribute to the loss-cone overfilling observed by ELFIN and reproduced in numerical simulations in Fig. 5 for electrons of 60–150 keV. Conversely, in Supplementary Fig. 7, loss-cone overfilling above 300 keV is absent. All plots with model data are obtained from theoretical equations provided in cited references.

## Data availability

THEMIS and ELFIN data used in this study are available in public repository at http://themis.ssl.berkeley.edu and https://data.elfin.ucla.edu/ela. ERG (Arase) data are available from the ERG Science Center operated by ISAS/JAXA and ISEE/Nagoya University (https://ergsc.isee.nagoya-u.ac.jp/index.shtml.en,[66]). The present study analyzed MEP-e-L2 data v01.01[67], and MGF-L2 data v04.04[68]. The source data used to produce figures in this study are publicly accessible at https://doi.org/10.6084/m9.figshare.19200305.v1. The datasets generated during and/or analyzed during the current study are available from the corresponding author on reasonable request.

## Code availability

Data analysis was done using Space Physics Environment Data Analysis Software (SPEDAS) V4.1, available at https://spedas.org/. The computer code of numerical simulations in this study is available upon request to the corresponding author.

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

## Acknowledgements

X.-J.Z., A.A., V.A., E.T., and C.W. acknowledge support by National Aeronautics and Space Administration (NASA) awards NNX14AN68G, 80NSSC20K1578, 80NSSC21K0729, NAS5-02099 and National Science Foundation (NSF) awards AGS-1242918, AGS-2019950. We are grateful to NASA's CubeSat Launch Initiative programme for successfully launching the ELFIN satellites in the desired orbits under ELaNa XVIII. We thank the Air Force Office of Scientific Research (AFOSR) for early support of the ELFIN programme under its University Nanosatellite Program, UNP-8 project, contract number FA9453-12-D-0285. We also thank the California Space Grant programme for student support during the project's inception. We acknowledge the hardware contributions and technical assistance of Mr. David Hinkley and The Aerospace Corporation. We acknowledge contributions of volunteers in the ELFIN team to the routine operations of the ELFIN mission. We thank Ms. Judith Hohl for text editing.

## Author contributions

X.-J.Z., A.A., V.A., D.M., S.K., and Y.M. discussed the concepts, X.-J.Z. performed data analysis of ELFIN and THEMIS, A.A. performed simulations, S.K. performed data analysis of ERG, D.M. made theoretical estimates for data/model comparison, E.T. and C.W. performed ELFIN EPD calibration and data preparation, S.K., S.Y., K.K., T.H., Y.M., I.S., and A.M. performed ERG calibration and data preparation, X.-J.Z., A.A., V.A., and D.M. wrote the manuscript.

## Competing interests

The authors declare no competing interests.
