## [Peer Review File · Nature Communications]

REVIEWER COMMENTS

Reviewer #1 (Remarks to the Author):

Review of "Superfast Precipitation of Energetic Electrons in Earth's Radiation Belts" by Zhang, Artemyev, Angelopoulos, Tsai, Wilkins, Kasahara, Mourenas, Yokota, Keika, Hori, Miyoshi, Shinohara and Matsuoka

The paper contains very interesting, new and exciting results. After suitable amendments, it should be publishable in Nature Communications.

Comments

Lines 41-43. It should also be mentioned here that quasi-linear theory has also been shown to not be applicable to whistler mode wave interactions with electrons because the waves have been shown to be monochromatic and coherent (JGRSP, 125, e2020JA028090, 2020, <https://doi.org/10.1029/2020JA028090>). The pitch angle scattering rates with coherent waves is ~1,000 times faster than with random phase emissions (JGR, 115, A00F15, 2010, doi:10.1029/2009JA014885). This faster rate of pitch angle scattering is the first time that electron microburst precipitation has been explained (JGRSP, 118, 1-17, 2013. doi:10.1002/jgra.50264).

Paragraph starting on line 48. The concept of strong diffusion will need to be revised with stronger pitch angle scattering rates associated with coherent waves. Since strong diffusion will be reached much much faster, is it possible that this is causing the energization and flux increase? Please discuss.

Line 113 paragraph. Same question as above.

Same paragraph. Does the code conserve energy? For example can a single wave accelerate an electron from 10 keV up to 200 keV as you state? I would think that the wave photon has far less energy than a 200 keV electron. Especially since these oblique waves have far less energy than the equatorial parallel propagating waves. If this is a problem is it a stochastic process involving interactions with many waves? The process might be slower however.

Please read and discuss the results of a very old auroral zone rocket experiment: JGR, 72, 23, 5817-5823, 1967. This experiment was done to understand the source of precipitating ~10 to 100 keV electrons. People were assuming that the source of scattering was at the magnetic equator where the wave amplitudes are the greatest. However the electron velocity dispersion results in this paper indicates a lower altitude source. This could of course be your oblique waves and Cerenkov acceleration. However 1967 was before double layers came into prominence. Please discuss electric fields in double layers as a possible source of causing more electrons to enter the loss cone. This might be another possible mechanism for your observations.

Reviewer #2 (Remarks to the Author):

In this manuscript the authors present low-altitude spacecraft measurements of loss-cone overfilling due to wave-particle resonant interactions. It is hypothesized that the acceleration of electrons and their transport into the loss-cone is due to the nonlinear Landau resonance with oblique whistler waves. This hypothesis is supported by conjugate high-altitude spacecraft observations near the magnetic equator. Long-term observations of loss-cone overfilling are compared with simulations of electron transport in energy–pitch-angle space due to Landau trapping and scattering.

I believe that the results could be significant and worth publishing in Nature Communications, however, there are several issues with the presented data and analysis which make it difficult to decide whether the conclusions of the paper are valid. These issues are described in the rest of this document. The list of minor issues (grammar, typos, proper referencing etc.) is not included, as it would not be very useful at this early stage of the review process.

Figures 2a-d show data from the spacecraft ELFIN-A, gathered approximately between 3:00 and 3:05 on Nov 23, 2020. During this 5-minute interval, the low-altitude satellite travels over a large range of L-shells (approx. 3-8 for dipole, 3.5-11 for TS04 model), covering essentially the entire outer radiation belt. During this time, the THEMIS-E satellite, which provides the wave field measurements, stays near the L-shell of 9.4 (or 9.1 for TS04). In Figures 2e-g, the THEMIS-E spacecraft position from 02:00 to approx. 03:30 is said to be in conjunction with ELFIN-A (delimited by two dash-dotted lines, with the left line being partially hidden behind the ticks on the vertical axis). I do not understand how could wave bursts with a lifetime of minutes, which were observed tens of minutes before or after the ELFIN-A observations, influence the particle precipitation. Also, the two satellites are at different local times (MLT), and it is not clear from the plotted data (latitude is not given, not even the hemisphere) whether the B_y component of the geomagnetic field is large enough to allow particles to travel from $MLT=12.5$ to $MLT=11.0$ within one half-bounce in the magnetic trap.

Continuing with Figure 3, the burst mode wave data shown there are from 2:28:35 – 2:28:47, which is again about half an hour before the event observed on ELFIN-A. Furthermore, the wave normal angles in Figure 3c seem to frequently reach values over 80 degrees (dark red color), which should typically not happen, because the resonance cone for the observed wave frequency and plasma parameters is only about 74 degrees. I was recently made aware by a colleague who works with THEMIS data that there is an (so far unpublished) issue with the search coils on THEMIS-E and D (increased noise floor, decreased sensitivity), which compromises the magnetic field data from about 2016-2017 onward. Therefore, the authors should consider searching for a conjunction event on a different spacecraft.

In Figure 4a-d, the times at which these data were taken are not given. Also, the MEP-e instrument on ERG has very large dead zones (according to the reference [26]), i.e. it captures particles only from a small section of each angular bin (and the bin is much larger than the typical loss cone size); I wonder how is this property of the instrument handled in the data analysis. The section Methods does not describe the ERG data analysis.

Around line 328, the authors describe the input parameters of their simulation (wave amplitude, frequency and wave normal angle) and refer to Figure S2c, which however shows phase space density and not wave parameters. And wave parameters in Figure S2d do not show the wave normal angle, which is instead simply set to be near the resonance cone (line 332). While the simulation is very robust and provides many interesting results, the manuscript provides no solid proof that the ELFIN observations are connected to very oblique waves, due to issues with time intervals and search coils described above. Actually, the most intense waves in Figure 2g around 3:30, which still fall into the annotated time interval, could very well be quasi-parallel chorus risers (burst mode is not available for these intense waves, so the actual character of these waves cannot be confirmed, but the emissions start at very low frequencies, and THEMIS observations show that highly oblique chorus almost never goes below $0.2 f/f_{ce}$ doi:10.1002/2014JA020575).

The phase space density in Figure S2c shows plateaus in both parallel and perpendicular section of the PSD distribution. It is not clear what exactly do the symbols for perpendicular and parallel direction represent, i.e. what range of pitch angles is covered in each direction. In any case, Landau resonance should produce a plateau in the reduced distribution $f(v_{parallel})$, and the existence of such plateau is not clear from the plots. Figure S2a shows a higher anisotropy measure for energies from 1 keV to 10 keV, but the shape of the full 2D distribution is still not obvious. Given the information presented in the plots, we could be observing a so-called shell distribution (a ring distribution where the enhancement is not only on 90 deg pitch angle, but at all pitch angles).

Concerning the Landau resonance hypothesis testing, not only is it not well supported by the provided Figures, it is also not well explained why there couldn't be other reasons for the loss cone overfilling at 100 keV energies. Notice that for the observed plasma parameters ($f_{pe}/f_{ce}=2$), the range of cyclotron resonant energies (with $v_{perp} = 0$) goes from 58 keV to 187 keV for wave frequencies from 0.3 to 0.15 of equatorial f_{ce} , and becomes even larger as the waves propagate away from the B-min. This energy range matches rather nicely with the energies observed on ELFIN. It was shown by Li et al. 2010 (doi:10.1029/2009JA014845) that the f_{pe}/f_{ce} ratio, which is an important contributor to the high cyclotron resonance energy, is low in the dawn–noon region, enabling the scattering of high energy electrons. Chorus waves with very low frequencies ($< 0.2 f_{ce}$) are observed preferentially at higher L-shells (doi:10.1002/2016GL067687, doi.org/10.1002/2016JA023561), which again agrees with the ELFIN results. ($J_{pr}/J_{tr} > 1$ might be connected to anomalous resonance or to some specific shape of the loss-cone distribution, but that

is a pure speculation from my side). Could this mean that the observed fluxes are due to cyclotron scattering of already accelerated electrons? The authors should briefly demonstrate that this alternate hypothesis, or any other viable hypothesis, cannot explain the observations better than the Landau resonance.

When it comes to the structuring of the manuscript, the main part of the text is almost wholly concerned with a case study (apart from a few sentences at the end), and the larger ELFIN statistics is discussed only within the Methods section. Even if the one thoroughly discussed case of very fast precipitation was indeed caused by Landau trapping, the paper does not give much explanation on why it should be the same for all the other cases. Conjunctions of long-term ELFIN measurements with THEMIS observation of oblique whistlers are not provided. This might not be a large issue if the case study presented in the main part was more convincing.

To summarize, there are some outstanding issues in the manuscript which the authors need to address in order to prove the validity of their methods and the significance of their results.

Response to reviewers' comments on *Superfast Precipitation of Energetic Electrons in Earth's Radiation Belts*

We are grateful to both Reviewers for their useful comments and suggestions. Our point-by-point responses to the comments and the related revisions in the manuscript are listed below in blue.

Response to the First Reviewer

The paper contains very interesting, new and exciting results. After suitable amendments, it should be publishable in Nature Communications.

- *Lines 41-43. It should also be mentioned here that quasi-linear theory has also been shown to not be applicable to whistler mode wave interactions with electrons because the waves have been shown to be monochromatic and coherent (JGRSP, 125, e2020JA028090, 2020, <https://doi.org/10.1029/2020JA028090>). The pitch angle scattering rates with coherent waves is $\sim 1,000$ times faster than with random phase emissions (JGR, 115, A00F15, 2010, [doi:10.1029/2009JA014885](https://doi.org/10.1029/2009JA014885)). This faster rate of pitch angle scattering is the first time that electron microburst precipitation has been explained (JGRSP, 118, 1-17, 2013. [doi:10.1002/jgra.50264](https://doi.org/10.1002/jgra.50264)).*

This is a good point! We have included further clarifications and references!

- *Paragraph starting on line 48. The concept of strong diffusion will need to be revised with stronger pitch angle scattering rates associated with coherent waves. Since strong diffusion will be reached much much faster, is it possible that this is causing the energization and flux increase? Please discuss.*

Line 113 paragraph. Same question as above.

This is an interesting question. The strong diffusion concept is based on the loss-cone size, the geometrical characteristic of the system. This concept implies that as diffusion does not increase local gradients in the phase space, but only smoothes such gradients, the diffusion-driven flux into the loss cone cannot exceed the flux lost from the loss cone due to the precipitation into the atmosphere. Thus, the increase of the diffusion rates, e.g., due to electron resonances with highly coherent waves, will allow the system to reach

the strong diffusion limit for smaller wave intensity, or equivalently for a smaller number of resonant interactions within one bounce period. But this likely does not change the fundamental concept of strong diffusion limit as the limit of a permanently full loss cone. Instead, this can only change the crucial wave intensity required to reach this limit.

- *Same paragraph. Does the code conserve energy? For example can a single wave accelerate an electron from 10 keV up to 200 keV as you state? I would think that the wave photon has far less energy than a 200 keV electron. Especially since these oblique waves have far less energy than the equatorial parallel propagating waves. If this is a problem is it a stochastic process involving interactions with many waves? The process might be slower however.*

The Reviewer is totally right, the wave energy is insufficient to accelerate any significant number of electrons during a single resonant interaction. However, the concept of nonlinear resonance does explain the acceleration of one electron population (trapped electrons) allowed by the energy lost by another electron population (bunched electrons). This concept has been introduced in (1) for Langmuir waves in inhomogeneous plasma, and then have been developed for whistler-mode waves in, e.g., (2; 3; 4). The idea is that for nonlinear wave-particle interactions, there are N_{trap} particles gaining energy as $N_{trap} \cdot (dE_{trap}/dt)$ and $N_{res} - N_{trap} \approx N_{res}$ particles losing energy as $-N_{res} \cdot (dE_{bunch}/dt)$, with the amount of decelerated particles much larger than the amount of accelerated/trapped particles. The ratio of accelerated to total resonant particles in the nonlinear resonance is $N_{trap}/N_{res} \sim \sqrt{B_w/B_0}$, but the ratio of energy gain rates are $(dE_{trap}/dt)/(dE_{bunch}/dt) \sim \sqrt{B_0/B_w}$. These equations explain how a small population of electrons can gain a large energy without wave damping: $N_{trap} \cdot (dE_{trap}/dt) - N_{res} \cdot (dE_{bunch}/dt) \approx N_{res}(dE_{trap}/dt) \cdot (N_{trap}/N_{res} - (dE_{bunch}/dE_{trap})) \sim 0$. This concept is intrinsically included into the nonlinear resonance theory, and in particular into our model, since in this model the calculations of number of trapped (accelerated) particles and rates of particle acceleration and deceleration scale with B_w in such a way that there is no energy change for the full population of all particles, but only an energy redistribution between particle sub-populations (5).

- *Please read and discuss the results of a very old auroral zone rocket experiment: JGR, 72, 23, 5817-5823, 1967. This experiment was done to*

understand the source of precipitating ~ 10 to 100 keV electrons. People were assuming that the source of scattering was at the magnetic equator where the wave amplitudes are the greatest. However the electron velocity dispersion results in this paper indicates a lower altitude source. This could of course be your oblique waves and Cerenkov acceleration. However 1967 was before double layers came into prominence. Please discuss electric fields in double layers as a possible source of causing more electrons to enter the loss cone. This might be another possible mechanism for your observations.

This 1967 JGR by Lampton provided interesting results on the temporal variability of 20-150 keV electron precipitation in the auroral zone as seen by detectors on rockets. However, we note that there was no loss cone overfilling in the measured pitch-angle distributions of their Figs. 3(a,b). Therefore, it is more likely that the precipitation could have been related to usual cyclotron resonance with chorus waves. However, electron detectors can be significantly contaminated by protons of 0.8-2 MeV, especially at high L as there. Therefore, it is hard to draw any clear conclusion from this paper in relation to our own study. Nevertheless, we added a brief discussion of double layers and Time Domain Structures effects in the Methods Section of our manuscript.

Response to the Second Reviewer

In this manuscript the authors present low-altitude spacecraft measurements of loss-cone overfilling due to wave-particle resonant interactions. It is hypothesized that the acceleration of electrons and their transport into the loss-cone is due to the nonlinear Landau resonance with oblique whistler waves. This hypothesis is supported by conjugate high-altitude spacecraft observations near the magnetic equator. Long-term observations of loss-cone overfilling are compared with simulations of electron transport in energy-pitch-angle space due to Landau trapping and scattering.

I believe that the results could be significant and worth publishing in Nature Communications, however, there are several issues with the presented data and analysis which make it difficult to decide whether the conclusions of the paper are valid. These issues are described in the rest of this document. The list of minor issues (grammar, typos, proper referencing etc.) is not included, as it would not be very useful at this early stage of the review process.

- *Figures 2a-d show data from the spacecraft ELFIN-A, gathered approximately between 3:00 and 3:05 on Nov 23, 2020. During this 5-minute interval, the low-altitude satellite travels over a large range of L-shells (approx. 3–8 for dipole, 3.5–11 for TS04 model), covering essentially the entire outer radiation belt. During this time, the THEMIS-E satellite, which provides the wave field measurements, stays near the L-shell of 9.4 (or 9.1 for TS04). In Figures 2e-g, the THEMIS-E spacecraft position from 02:00 to approx. 03:30 is said to be in conjunction with ELFIN-A (delimited by two dash-dotted lines, with the left line being partially hidden behind the ticks on the vertical axis). I do not understand how could wave bursts with a lifetime of minutes, which were observed tens of minutes before or after the ELFIN-A observations, influence the particle precipitation. Also, the two satellites are at different local times (MLT), and it is not clear from the plotted data (latitude is not given, not even the hemisphere) whether the B_y component of the geomagnetic field is large enough to allow particles to travel from $MLT = 12.5$ to $MLT = 11.0$ within one half-bounce in the magnetic trap.*

The Reviewer is right, we did not use the term **conjunction** correctly. We do not mean there is one-to-one correlation between low-altitude measurements of precipitating electrons and equatorial measurements of whistler-mode waves. We have included additional justifications in the main text. Our idea is that ELFIN continuously measured bursts of intense precipitation at $L > 6$, and the most intense precipitation shows $j_{prec}/j_{trap} > 1$ features, whereas THEMIS measurements show that this entire L -shell range is continuously filled by intense, very oblique whistler-mode waves, from one hour before up to one hour after ELFIN observations. A THEMIS statistics has shown that the typical correlation length of the source region of whistler-mode waves is ~ 1.4 hours in MLT and 1.5 Earth radius radially at $L = 7.5 - 10$ near 11-12 MLT (6). Accordingly, an approximate conjunction (within 1.4 MLT, 1.5 in L , and 35 min UT) between ELFIN precipitation and THEMIS wave source region occurred between 02:26 UT and 03:16 UT in Fig. 2. Therefore, it is likely that the very oblique whistler-mode waves observed by THEMIS at $L \sim 7.5 - 10$ and 12.0-12.7 MLT around 03:00 UT (and near 02:28 UT in Fig. 3) drove the precipitation measured by ELFIN at $L = 7.5 - 10$ and 11.2 MLT near 3:02 UT. It is reasonable to assume that such very oblique whistler-mode waves resonated with electrons and provided strong (with $j_{prec}/j_{trap} > 1$) precipitation. The numerical simulation confirms this assumption. Thus, our conclusions are not based on a one-to-one conjunction of equatorial and low-altitude spacecraft measurements (it

is almost impossible to get such a perfect conjunction due to the very fast motion of low-altitude spacecraft), but rather on a near-conjunction based on the measured duration and statistical correlation lengths of the source region of whistler-mode waves, and on simulations driven by equatorial measurements that reproduce well the precipitation of the electron pitch-angle distributions measured by ELFIN.

- *Continuing with Figure 3, the burst mode wave data shown there are from 2:28:35 – 2:28:47, which is again about half an hour before the event observed on ELFIN-A. Furthermore, the wave normal angles in Figure 3c seem to frequently reach values over 80 degrees (dark red color), which should typically not happen, because the resonance cone for the observed wave frequency and plasma parameters is only about 74 degrees. I was recently made aware by a colleague who works with THEMIS data that there is an (so far unpublished) issue with the search coils on THEMIS-E and D (increased noise floor, decreased sensitivity), which compromises the magnetic field data from about 2016-2017 onward. Therefore, the authors should consider searching for a conjunction event on a different spacecraft.*

Thank you for raising this important comment! We checked and indeed found significant problems with SCM measurements during this time interval. After communicating with the SCM P.I., we decided to exclude SCM data from the analysis and to focus on electric field measurements that also show very oblique wave propagation. The Reviewer is right, the wave burst measurements are performed in the same region 33 minutes before ELFIN measurements and, thus, we can only derive the general information about whistler-mode waves in this event, but cannot use a perfect conjunction and instead base our analysis on a near-conjunction between ELFIN precipitation and THEMIS’s wave source region (see response to your first comment). Thus, we use electric field and cold plasma dispersion to estimate the wave normal angle for our simulation. Although these are relatively uncertain estimates, electric field shows that waves are propagating within $[5, 10]^\circ$ from the resonance cone angle, and this information is sufficient to construct the model of wave-particle interaction. We use the waves distributed between $[\theta_r - 10^\circ, \theta_r - 5^\circ]$ for our simulations; this is also consistent with previous statistics of wave-normal angles for very oblique whistler-mode waves (7). We also replaced SCM measurements by electric field measurements and recalculated the distribution of wave amplitudes and frequencies for our simulations.

- In Figure 4a-d, the times at which these data were taken are not given. Also, the MEP-e instrument on ERG has very large dead zones (according to the reference [26]), i.e. it captures particles only from a small section of each angular bin (and the bin is much larger than the typical loss cone size); I wonder how is this property of the instrument handled in the data analysis. The section Methods does not describe the ERG data analysis.

Times for ELFIN data and for ERG data have been added to the caption. Regarding ERG measurements: since there are dead zones in the 4π steradian solid angle, MEP-e sees the loss cones sometimes and misses them at other times. The precipitating flux j_{prec} in Fig. 4 is calculated based on the electron count during the time period when one of the MEP-e scopes captures the loss cone (time step = 15.6 ms for an energy bin). The count rate is then converted to the differential flux by being divided by the geometric factor and energy and multiplied by a sensitivity correction factor (see the reference (8)). For the time period in which MEP-e does not cover the loss cones, we cannot acquire j_{prec} .

For the trapped flux j_{trap} , here we use the pitch angle range of 5 – 15 deg. Generally speaking this pitch angle range can be scanned multiple times by multiple telescopes during a spacecraft spin (= 8 s). We take averages of these multiple flux values in every spacecraft spin. We have added a summary of these details into the Method.

- Around line 328, the authors describe the input parameters of their simulation (wave amplitude, frequency and wave normal angle) and refer to Figure S2c, which however shows phase space density and not wave parameters. And wave parameters in Figure S2d do not show the wave normal angle, which is instead simply set to be near the resonance cone (line 332). While the simulation is very robust and provides many interesting results, the manuscript provides no solid proof that the ELFIN observations are connected to very oblique waves, due to issues with time intervals and search coils described above. Actually, the most intense waves in Figure 2g around 3:30, which still fall into the annotated time interval, could very well be quasi-parallel chorus risers (burst mode is not available for these intense waves, so the actual character of these waves cannot be confirmed, but the emissions start at very low frequencies, and THEMIS observations show that highly oblique chorus almost never goes below $0.2f/f_{ce}$ doi:10.1002/2014JA020575).

Sorry for these misprints, they have been corrected. The Reviewer is right, the main wave characteristics derived from observations are the wave

amplitude and frequency, whereas the wave normal angle θ is simply set to be large (i.e., very oblique whistler-mode waves are considered). This simplification is because we do not have a perfect conjunction between THEMIS wave measurements and ELFIN observations (as mentioned by the Reviewer). Moreover, models of wave-particle interactions do not depend strongly on specific values of the wave normal angle, but are determined by one of two wave modes: quasi-parallel whistler mode waves with θ below Gendrin angle or very oblique whistler-mode waves with θ near the resonance cone angle (9; 10). These two wave modes are usually well separated in observations (7; 11) and there are some physical reasons explaining why they cannot co-exist (12). This is why we use THEMIS measurements to determine which wave mode (very oblique or quasi-parallel) was observed within the entire interval of near-conjunction with ELFIN precipitation and then use this wave mode in our simulations.

Responding to one of the next Reviewer’s questions, we now discuss in the Methods Section of the paper why quasi-parallel waves resonating with electrons in cyclotron resonance may not explain the observed loss-cone overfilling. Moreover, we provide an additional dataset in the Supplementary Information (Supplementary Fig. 2), to show that such loss-cone overfilling observations are indeed often associated with very oblique whistler-mode wave observations.

Although (13) found nearly no very-oblique chorus waves at $f < 0.2f_{ce}$ in their selected rising or falling tone chorus elements using 2008 THEMIS data, it is worth noting that a much larger statistics from (7) based on 2012-2015 data from the Van Allen Probes, has shown the presence of a lot of very-oblique lower-band chorus waves less than 10° away from the resonance cone angle at $f = (0.13-0.20) \times f_{ce}$ (at the same frequencies as many waves seen at 02:45-03:16 UT in our Fig. 2), with only ~ 10 times smaller occurrences than at $f = 0.3f_{ce}$ (see their Fig. 3b and Fig. 5a). (14) also found very-oblique lower band chorus waves at $L = 7.5 - 9.0$ and 11-12 MLT (in the same region as in our Fig. 2) using THEMIS data. Within our near-conjunction interval from 02:26 UT to 03:16 UT in Fig. 2, we clearly identified near 02:28 UT in Fig. 3 very oblique waves with $\theta \in [\theta_r - 10^\circ, \theta_r - 5^\circ]$ at $(0.20 - 0.25)f_{ce}$, and similar waves are seen from 02:26 UT to 03:16 UT in Fig. 2. As we now underline in lines 270-273: "Although Supplementary Fig. 3d contains slightly lower f/f_{ce} values than Figure 3, note that the Landau resonant parallel electron energy $\mathcal{E}_{res}[\text{keV}] \simeq 250(f_{ce}/f_{pe})^2(1 - \cos\theta_r/\cos\theta)\cos\theta_r/\cos\theta$, most important in simulations, is nearly independent of f/f_{ce} for $0.14 < f/f_{ce} < 0.25$

in this θ range” (i.e., for $\theta \in [\theta_r - 10^\circ, \theta_r - 5^\circ]$).

- *The phase space density in Figure S2c shows plateaus in both parallel and perpendicular section of the PSD distribution. It is not clear what exactly do the symbols for perpendicular and parallel direction represent, i.e. what range of pitch angles is covered in each direction. In any case, Landau resonance should produce a plateau in the reduced distribution $f(v_{\parallel})$, and the existence of such plateau is not clear from the plots. Figure S2a shows a higher anisotropy measure for energies from 1 keV to 10 keV, but the shape of the full 2D distribution is still not obvious. Given the information presented in the plots, we could be observing a so-called shell distribution (a ring distribution where the enhancement is not only on 90 deg pitch angle, but at all pitch angles).*

The Reviewer is right, the phase space density demonstrates the presence of a plateau in both parallel ($\alpha < 30^\circ$) and perpendicular ($\alpha \in [75, 105]^\circ$) directions. However, we did not suggest that the formation of such a plateau is due to the Landau resonance. Previous estimates suggest that whistler-mode waves cannot form such a plateau by themselves (15; 16). The main idea here is that such electron distribution functions with reduced $\partial f/\partial v_{\parallel}$ (or even $\partial f/\partial v$) represent favorable conditions for the generation of very-oblique whistler-mode waves (17; 18), which cannot be generated in the presence of a strong negative $\partial f/\partial v_{\parallel}$. We have added this clarification into the text. The mechanisms of formation of such electron distribution functions with reduced $\partial f/\partial v_{\parallel}$ should be related to more global processes (enhanced convection, strong ionosphere outflow, etc.) than wave-particle interactions.

- *Concerning the Landau resonance hypothesis testing, not only is it not well supported by the provided Figures, it is also not well explained why there couldn't be other reasons for the loss cone overfilling at 100 keV energies. Notice that for the observed plasma parameters ($f_{pe}/f_{ce} = 2$), the range of cyclotron resonant energies (with $v_{\perp} = 0$) goes from 58 keV to 187 keV for wave frequencies from 0.3 to 0.15 of equatorial f_{ce} , and becomes even larger as the waves propagate away from the B-min. This energy range matches rather nicely with the energies observed on ELFEN. It was shown by Li et al. 2010 (doi:10.1029/2009JA014845) that the f_{pe}/f_{ce} ratio, which is an important contributor to the high cyclotron resonance energy, is low in the dawn-noon region, enabling the scattering of high energy electrons. Chorus waves with very low frequencies ($< 0.2f_{ce}$) are observed preferentially at higher L-shells*

(doi:10.1002/2016GL067687, doi.org/10.1002/2016JA023561), which again agrees with the ELFIN results. ($J_{pr}/J_{tr} > 1$ might be connected to anomalous resonance or to some specific shape of the loss-cone distribution, but that is a pure speculation from my side). Could this mean that the observed fluxes are due to cyclotron scattering of already accelerated electrons? The authors should briefly demonstrate that this alternate hypothesis, or any other viable hypothesis, cannot explain the observations better than the Landau resonance.

Thank you for raising this important question about alternative mechanisms of observed loss-cone overfilling. We initially studied various alternative possibilities, but did not include all this discussion into the text, due to size limits. There are two main resonances (Landau and cyclotron) and three types of interactions (diffusion, phase bunching, phase trapping). Let us consider below all six possible combinations. For simplicity of illustration, we will consider a monochromatic wave, but the same idea works for rising/falling tone chorus waves. The Figure below shows resonance curves in (energy, pitch-angle) space, and wave-particle resonant interactions move electrons along such curves (3): $(\gamma^2 - 1) \sin^2 \alpha_{eq} = const$ for Landau resonance (conservation of the magnetic moment) and $(2\Omega_{ce,eq}/\omega)\gamma - (\gamma^2 - 1) \sin^2 \alpha_{eq} = const$ for cyclotron resonance. The cyclotron resonance curves are quite flat around the loss cone for energies ~ 100 keV, i.e., cyclotron resonance moves particles into the loss cone with a small energy loss. In this case, diffusion and phase bunching (small random energy change or directed energy decrease) will move particles gradually toward the loss cone following the gradient of initial distribution $\partial f / \partial \alpha_{eq} > 0$. These processes tend to smooth the gradients and will not create any new gradients in pitch-angle space, because resonance curves are quite flat. The cyclotron phase trapping may change electron energy significantly and create a local gradient in the (energy, pitch-angle) space (see figure below for typical energy changes due to trapping and bunching), but this trapping increases electron pitch angle and move particles away from the loss cone. The specific regime of anomalous cyclotron trapping (19; 20; 21) will provide an effective electron transport away from the loss cone, but this regime does not assume any particle flow into the loss cone. One can suggest the scenario when anomalous trapping evacuate electrons from the vicinity of the loss cone, whereas phase bunching moves electrons directly into the loss cone (creating inverse gradient $\partial f / \partial \alpha_{eq} < 0$), but this scenario does not work for realistic wave characteristics: the pitch-angle (magnetic moment) range of anomalous trapping is about the pitch-angle range of phase bunching, and all electrons evacuated from the loss-cone vicinity by anomalous trapping will

be replaced by even more numerous electrons scattered into this pitch-angle range by bunching (see Discussion in (21)). Thus, cyclotron resonance cannot create $\partial f/\partial\alpha_{eq} < 0$ within the loss cone, i.e., cannot create the loss-cone overfilling. The only way to get $J_{precip} > J_{trapped}$ via anomalous trapping would be to have a strong atmospheric backscatter in the opposite hemisphere, sending an initially flat pitch-angle spectrum (with $J_{precip} = J_{trapped}$) inside the loss cone toward ELFIN, and next, anomalous trapping near the equator that would slightly increase J_{precip} versus $J_{trapped}$ before such electrons reach ELFIN. However, this is not possible because atmospheric backscatter is not sufficiently efficient: it leads to a small $J_{precip}(\alpha < 0.5\alpha_{LC}) < 0.3 J_{trapped}$ at 100-200 keV (22). Moreover, anomalous trapping is stronger at lower energy (19; 20; 21), which should produce a stronger peak of flux $J_{precip} > J_{trapped}$ at lower energy, contrary to the present ELFIN observations.

Let us consider Landau resonance (see figure below for typical energy changes due to trapping and bunching in Landau resonance). The main feature of the Landau resonance is that resonance curves have very strong gradients around the loss cone, i.e., all electrons moving to the small pitch angles should gain energy. Moreover, in contrast to the cyclotron resonance, the Landau trapping leads to an electron pitch-angle decrease. Thus, trapping can move low-energy (associated with the large phase space density) electrons into the loss cone from moderate pitch angles, and this transport will be associated with an energy gain. This mechanism may explain formation of $\partial f/\partial\alpha_{eq} < 0$ gradients during the short interval of a bursty precipitation.

We have included a short version of this discussion into the text. Since cyclotron resonance with high or low frequency waves cannot produce the observed loss-cone overfilling, we only briefly mentioned the presence of low-frequency chorus waves at high L in the dawn-noon sector of lower $f_{pe}/f_{ce} \sim 5$, which leads to minimum energies of ~ 40 keV for cyclotron resonance, with the suggested references.

- *When it comes to the structuring of the manuscript, the main part of the text is almost wholly concerned with a case study (apart from a few sentences at the end), and the larger ELFIN statistics is discussed only within the Methods section. Even if the one thoroughly discussed case of very fast precipitation was indeed caused by Landau trapping, the paper does not give much explanation on why it should be the same for all the other cases. Conjunctions of long-term ELFIN measurements with THEMIS observation of oblique whistlers are not provided. This might not be a large issue if the case*

Fig. 1: Left panel shows the resonance curves for cyclotron and Landau resonances (directions of phase trapping and bunching are shown). Middle and right panels show typical energy and pitch-angle changes for cyclotron and Landau nonlinear resonances.

study presented in the main part was more convincing.

The Reviewer is right, taking into account the general uncertainties of any comparisons of near-equatorial and low-altitude spacecraft, additional confirmation about the statistics of the loss-cone overfilling had to be provided. Since some of the events in the statistics shown in the Supplementary Fig. 1 had conjugate ELF/IN/THEMIS measurements, we picked up three of these events to show (in Supplementary Fig. 2) that the loss-cone overfilling phenomenon is indeed associated with equatorial observations of very oblique (large parallel electric field, non circular polarization) whistler-mode waves in a number of events (and not only in one event).

References

- [1] Karpman V I, Istomin J N and Shklyar D R 1975 *Physics Letters A* **53** 101–102
- [2] Omura Y, Katoh Y and Summers D 2008 **113** A04223
- [3] Shklyar D R and Matsumoto H 2009 *Surveys in Geophysics* **30** 55–104
- [4] Shklyar D R 2011 *Annales Geophysicae* **29** 1179–1188

- [5] Artemyev A V, Neishtadt A I, Vasiliev A A and Mourenas D 2016 *Physics of Plasmas* **23** 090701
- [6] Agapitov O V, Mourenas D, Artemyev A, Mozer F S, Bonnell J W, Angelopoulos V, Shastun V and Krasnoselskikh V 2018
- [7] Li W, Santolik O, Bortnik J, Thorne R M, Kletzing C A, Kurth W S and Hospodarsky G B 2016 **43** 4725–4735
- [8] Kasahara S, Yokota S, Mitani T, Asamura K, Hirahara M, Shibano Y and Takashima T 2018 *Earth, Planets, and Space* **70** 69
- [9] Mourenas D, Artemyev A V, Ripoll J F, Agapitov O V and Krasnoselskikh V V 2012 **117** A06234
- [10] Artemyev A V, Mourenas D, Agapitov O V and Krasnoselskikh V V 2013 *Annales Geophysicae* **31** 599–624
- [11] Santolík O, Macúšová E, Kolmašová I, Cornilleau-Wehrlin N and Conchy Y 2014 **41** 2729–2737
- [12] Agapitov O V, Mourenas D, Artemyev A V and Mozer F S 2016 **43** 11,112–11,120 ISSN 1944-8007 URL <http://dx.doi.org/10.1002/2016GL071250>
- [13] Taubenschuss U, Khotyaintsev Y V, Santolík O, Vaivads A, Cully C M, Contel O L and Angelopoulos V 2014 **119** 9567–9578
- [14] Gao X, Mourenas D, Li W, Artemyev A V, Lu Q, Tao X and Wang S 2016 *Journal of Geophysical Research (Space Physics)* **121** 6732–6748
- [15] Chen R, Gao X, Lu Q and Wang S 2019 *Geophysical Research Letters* (Preprint <https://agupubs.onlinelibrary.wiley.com/doi/pdf/10.1029/2019GL085108>) URL <https://agupubs.onlinelibrary.wiley.com/doi/abs/10.1029/2019GL085108>
- [16] Artemyev A V and Mourenas D 2020 *Journal of Geophysical Research (Space Physics)* **125** e27735
- [17] Li W, Mourenas D, Artemyev A V, Bortnik J, Thorne R M, Kletzing C A, Kurth W S, Hospodarsky G B, Reeves G D, Funsten H O and Spence H E 2016 **43** 8867–8875

- [18] Artemyev A V, Agapitov O, Mourenas D, Krasnoselskikh V, Shastun V and Mozer F 2016 **200** 261–355
- [19] Kitahara M and Katoh Y 2019 **124** 5568–5583
- [20] Albert J M, Artemyev A V, Li W, Gan L and Ma Q 2021 *Journal of Geophysical Research: Space Physics* **126** e2021JA029216
- [21] Artemyev A V, Neishtadt A I, Albert J M, Gan L, Li W and Ma Q 2021 *Physics of Plasmas* **28** 052902
- [22] Marshall R A and Bortnik J 2018 **123** 2412–2423

REVIEWER COMMENTS

Reviewer #1 (Remarks to the Author):

The authors have answered my comments. I approve of publication of the revised version of this paper.

Reviewer #2 (Remarks to the Author):

Review of "Superfast Precipitation of Energetic Electrons in Earth's Radiation Belts" by Zhang et al.,
Revision No. 1

The manuscript has been revised according to the suggestions and comments from the previous review. The uncalibrated B-field data from THEMIS have been removed and replaced by E-field measurements with B-field inferred from the cold plasma dispersion relation. The treatment of the approximate conjunction between THEMIS and ELFIN is discussed, and it is now clearer in which sense can the spacecraft data support the theoretical explanation for loss-cone overfilling. The discussion of various resonance processes has been expanded, making now a convincing case for Landau trapping. All these additions are accompanied by appropriate references.

To summarize, the manuscript presents new and significant experimental results on rapid electron precipitation into the atmosphere, complemented by a well-argued hypothesis on the origin of the large electron fluxes. Nevertheless, there are still some potential minor issues that I would like to see addressed or discussed:

Line 42-44: I am not sure if quasi-linear theory simply underestimates the jumps in phase space. It rather completely overlooks the highly organized motion of trapped electrons in wide resonance islands. Also, based on the literature known to me, it does not seem that interaction with large-amplitude whistler waves results in simple diffusion. See, e.g., Allanson et al. 2021 (DOI: 10.1029/2020JA028793), who suggest that advection should be taken into account.

Line 298-300: If I understand the implementation of wave properties into the mapping technique correctly, it seems that any short-time correlations are excluded. By correlations, I mean, e.g., the appearance of near-monotonous sequences of frequency samples with high amplitudes, which is characteristic of chorus elements. Do you think such correlations could affect your simulation results?

Line 162: There seems to be a missing verb in “they can only < 20 keV precipitating fluxes from this effect”.

Fig 2,4: Some text labels overlap with ticks and plotted lines. (This can be possibly resolved later, after the manuscript is accepted.)

Fig S4e: Part of the colour bar is missing (3.75-4.25).

Response to reviewers' comments on *Superfast Precipitation of Energetic Electrons in Earth's Radiation Belts*

We are grateful to both Reviewers for their positive responses and to the second Reviewer for helpful comments. Our point-by-point responses to the comments and the related revisions in the manuscript are listed below in blue.

Response to the First Reviewer

The authors have answered my comments. I approve of publication of the revised version of this paper.

We are thankful to the Reviewer for the positive evaluation of the revised manuscript.

Response to the Second Reviewer

The manuscript has been revised according to the suggestions and comments from the previous review. The uncalibrated B-field data from THEMIS have been removed and replaced by E-field measurements with B-field inferred from the cold plasma dispersion relation. The treatment of the approximate conjunction between THEMIS and ELFEN is discussed, and it is now clearer in which sense can the spacecraft data support the theoretical explanation for loss-cone overfilling. The discussion of various resonance processes has been expanded, making now a convincing case for Landau trapping. All these additions are accompanied by appropriate references.

To summarize, the manuscript presents new and significant experimental results on rapid electron precipitation into the atmosphere, complemented by a well-argued hypothesis on the origin of the large electron fluxes. Nevertheless, there are still some potential minor issues that I would like to see addressed or discussed:

- *Line 42-44: I am not sure if quasi-linear theory simply underestimates the jumps in phase space. It rather completely overlooks the highly organized*

motion of trapped electrons in wide resonance islands. Also, based on the literature known to me, it does not seem that interaction with large-amplitude whistler waves results in simple diffusion. See, e.g., Allanson et al. 2021 (DOI: 10.1029/2020JA028793), who suggest that advection should be taken into account.

We apologize for this inaccurate sentence. This sentence has been revised and a reference to Allanson et al., 2021 has been added.

- *Line 298-300: If I understand the implementation of wave properties into the mapping technique correctly, it seems that any short-time correlations are excluded. By correlations, I mean, e.g., the appearance of near-monotonous sequences of frequency samples with high amplitudes, which is characteristic of chorus elements. Do you think such correlations could affect your simulation results?*

Thank you for raising this good point! Indeed, we consider each wave packet separately and do not include any correlations between two successive wave packets/two successive resonances. Although such correlations may modify the nonlinear interactions and make them more effective (e.g., by providing several successive trappings, see (1)), our previous statistical analysis of intense chorus wave packets has shown that there are usually significant internal fluctuations of wave properties (frequency, phase) inside a chorus element, sufficiently strong to destroy such correlations between successive wave packets (see 2; 3). As written in the revised manuscript (lines 273 and 284-285), the adopted mapping technique is based on the cited Ref.30 (4), where this technique and its limitations have been extensively discussed, and where the importance of frequency/phase fluctuations between wave packets have been mentioned (p.20).

- *Line 162: There seems to be a missing verb in “they can only < 20 keV precipitating fluxes from this effect”.*

Thank you for catching this! It has been corrected.

- *Fig 2,4: Some text labels overlap with ticks and plotted lines. (This can be possibly resolved later, after the manuscript is accepted.)*

These have been fixed accordingly. Thank you for the reminder!

- *Fig S4e: Part of the colour bar is missing (3.75-4.25).*

This has been fixed. Thank you for catching it!

References

- [1] Hiraga R and Omura Y 2020 *Earth, Planets, and Space* **72** 21
- [2] Zhang X J, Mourenas D, Artemyev A V, Angelopoulos V, Kurth W S, Kletzing C A and Hospodarsky G B 2020 **47** e88853
- [3] Zhang X J, Agapitov O, Artemyev A V, Mourenas D, Angelopoulos V, Kurth W S, Bonnell J W and Hospodarsky G B 2020 **47** e89807
- [4] Artemyev A V, Neishtadt A I, Vasiliev A A, Zhang X J, Mourenas D and Vainchtein D 2021 *Journal of Plasma Physics* **87** 835870201 (*Preprint* 2011.00208)

REVIEWERS' COMMENTS

Reviewer #2 (Remarks to the Author):

The authors have satisfactorily answered my additional questions and made appropriate minor corrections to the manuscript. I recommend the current version of the manuscript for publication in Nature Communications.